# A Stope Mining Design with Consideration of Hanging Wall When Transitioning from Open Pit Mining to Underground Mining for Sepon Gold Mine Deposit, Laos

**Seelae Phaisopha** [1,2,*], **Hideki Shimada** [1], **Takashi Sasaoka** [1], **Akihiro Hamanaka** [1], **Phanthoudeth Pongpanya** [3], **Seva Shorin** [4] and **Khounma Senthavisouk** [4]

1   Department of Earth Resources Engineering, Kyushu University, Fukuoka 819-0395, Japan; shimada@mine.kyushu-u.ac.jp (H.S.); sasaoka@mine.kyushu-u.ac.jp (T.S.); hamanaka@mine.kyushu-u.ac.jp (A.H.)
2   Department of Mining Engineering, Polytechnic College, Vientiane 0100, Laos
3   Department of Mining Engineering, Faculty of Engineering, National University of Laos, Vientiane 0100, Laos; phanthoudeth.p@gmail.com
4   Lane Xang Minerals Limited, Vientiane 4486, Laos; seva.shorin@lxml.la (S.S.); khounma.senthavisouk@lxml.la (K.S.)
*   Correspondence: seelaepsp@gmail.com

**Abstract:** This study investigates the transition from surface to underground mining at the Sepon Gold mine. The stability of surface slopes is assessed prior to commencing underground operations. Stope mining is adopted based on ore body characteristics and geological features. Finite element numerical analysis, employing the Generalized Hoek–Brown criterion, evaluates slope stability for surface slopes and opening stopes. The pit design comprises a 70° slope angle, 20 m height, and 10–15 m safety berm, meeting stability requirements with a factor of safety of 2.46. Underground mining design includes a 62° ore body dip, a 50 m thick crown pillar to prevent surface subsidence, and 5 m wide, 5 m high stopes. Sill pillars are left for support after each level of excavation. Rock bolts are employed in specific stope areas where necessary, with continuous monitoring for surface deformation. This study analyzes the influence of stope sizes on the pit wall and pit bottom stability, identifying slope failures near the hanging wall close to the pit bottom during underground mining. A slight increase in the displacement zone is observed on the upper crest of the top footwall. Overall, the findings demonstrate successful stability in the transition from surface to underground mining at the Sepon Gold mine.

**Keywords:** open pit slope stability; finite element analysis; generalized Hoek–Brown criterion; underground stopping stability; surface subsidence



## 1. Introduction

To continue mining operations and to obtain greater productivity, a transition from surface mining to underground mining is necessary, as the costs associated with waste removal and management experience a significant exponential increase [1]. Bakhtavar [2] determined the optimal transition depth from open pit to underground mining in cases where combined mining methods are feasible. A proposed methodology utilizes economic block models of open-pit and underground methods, considering their respective Net Present Values (NPVs). They also focused on optimizing the transition from open-pit to underground mining for near-surface deposits with a vertical extent [3]. It employs (0–1) integer programming to maximize profits and make informed decisions regarding the transition depth. The model incorporates the block economic value of open-pit and underground mining. A hypothetical example is used to evaluate the effectiveness of the model. Canadian open-stope mining is the primary method used in underground, hard-rock mines. It involves small, fast-turnaround stopes (20,000 to 100,000 t) with poor to fair rock mass

conditions. Stability is ensured through stope dimensions, cable bolting, and diligent dilution control. The orebodies are typically deep, irregularly shaped, and high-grade. Pillarless or early pillar recovery strategies prevent overstressing, and stopes are filled, often with a cemented fill. This study highlights the commonalities in Canadian open-stope mines [4]. Chung [5] proposed a mixed-integer programming model for optimizing and planning shallow deposits in mines that can be extracted using both open-pit and underground methods. The model aims to find the best transition point and period, taking into account factors such as crown pillar placement and development cost, to maximize the net present value of the project. Through a case study, the optimal transition points and periods were generated for various scenarios. These findings indicate that the transition points range from 315 to 360 m below the surface. However, the transition from open-pit to underground mining presents challenges in terms of geotechnical aspects, infrastructure definition, and meeting project deadlines and production targets [6]. Eberhardt [7] conducted an implementation that addressed the challenges and risks associated with transitioning from surface to underground mass mining. It employs remote sensing, numerical modeling, and monitoring technologies to study rock engineering interactions and surface deformations caused by cave propagation. These findings improve our understanding of mass mining-induced subsidence and enhance subsidence prediction.

The economic decision to optimize the extraction of a mineral deposit involves considering both surface mining methods and underground (UG) mining methods to determine the most cost-effective mining option for the deposit. In resource development planning, the optimization of resource exploitation heavily relies on the chosen mining option for extraction. The term "mining options optimization" is used by researchers and professionals in the extractive industry to describe the initiatives or choices made to expand, modify, postpone, abandon, or adopt strategies for mining method(s) and occasionally investment opportunities. These decisions are based on changing economics, technology, or market conditions [3,8–11].

Recently, authors in mining engineering have commonly pre-selected a transition depth, known as the crown pillar, to evaluate mineral deposits. This division allows separate assessments of the area above the crown pillar for open pit mining and below for underground mining. By considering geological factors, economics, and other relevant considerations, the authors aim to optimize extraction by employing the most suitable mining methods for each portion [12–14]. Pre-selecting the crown pillar depth may not always lead to optimal solutions. To address this, authors have incorporated crown pillar positioning in the optimization process, evaluating multiple locations to achieve better outcomes for mineral extraction [3,15–18]. While the models developed by these authors represented significant improvements over previous works, they had certain limitations. Notably, they did not account for essential constraints such as ventilation requirements and the practical implementation of rock strength properties. The transition from open pit (OP) to underground (UG) mining involves a complex geomechanical process that necessitates careful consideration of various rock mass properties to ensure a successful and sustainable transition [19,20].

Afum and Ben-Awuah [21] applied a strategic mine plan aiming to optimize resource utilization, sustain mining profitability, and ensure a continuous supply of quality ore. It involves integrating surface and underground mining options and their interactions. Understanding existing tools and methodologies for transition planning is crucial for complex deposits amenable to open-pit and underground mining. MacNeil and Dimitrakopoulos [22] proposed an approach to determining the optimal depth for transitioning from open pit to underground mining, considering technical risk management. The method calculates the value of different transition depths by optimizing production schedules and cash flow projections. It integrates geological uncertainty using a stochastic program, leading to a 9% increase in net present value compared to a deterministic approach. The approach reduces mining project vulnerability to geological risk and improves decision-making by considering uncertainties in a three-dimensional context.

To address these concerns, we conducted a parametric study to investigate the design and influence of the stope, not only in the vicinity of the excavation but also in relation to open-pit displacement, particularly at the pit bottom and pit walls in the Sepon gold mine. According to the classification in the previous report [23], the transition scenario can be categorized into three types. The first involves transitioning from open-pit mining to underground mining, where both operations are carried out simultaneously. The second type involves underground mining below or adjacent to an existing open pit, indicating that open-pit mining has ceased in that area. The third type involves open-pit mining that extends through existing underground workings, often expanding the scale of mining to include high-grade zones and reopening closed historic underground mines. In the context of these three scenarios, the second type is specifically relevant and has been implemented in the case of Sepon gold mine deposits. In 1988, Potvin identified three fundamental aspects of engineering rock mechanics for the design of open stopes. These aspects include the characteristics of the rock mass; the effects of stress fields on the rock mass; and physical conditions determined by the size, geometry, and orientation of the rock mass and stress field. There are three main types of opening stopes that differ from those of other underground extraction methods. The first type is an opening stope with a non-entry method, followed by stopes that remain open until the final dimensions are achieved. Finally, underground excavations are designed for stability rather than caving methods [24]. Among these, the second type is utilized in the Sepon gold mine deposit. Moreover, in underground mining operations, the primary consideration is the impact of stress on an open stope. Stress relaxation in a rock mass is a time-dependent phenomenon characterized by the reduction in stress at a constant strain, which leads to the deformation or weakening of the rock mass over time [25]. This study investigates the transition from surface to underground mining at the Sepon Gold mine.

Moreover, a comprehensive study was conducted on the challenging issue of transitioning from surface mining to underground open-stope mining at the Sepon Gold mine. It involved a literature review focusing on the stability of the surface slope before commencing underground operations. The study area's geological features and the orebody's dip posed significant challenges. By reviewing relevant literature, including case studies and technical reports, valuable insights were gained to understand the complexities involved in this transition. This study aimed to ensure the safety and stability of the mining operations by informing decision-making and implementing effective strategies.

## 2. Geology of Gold and Copper Mineralization at Sepon Mineral District

### 2.1. Sepon Mineral District

Sepon is located in the Vilabouly district, Savannakhet province, in the south-central part of Laos. The Sepon project comprises 1947 km$^2$, located approximately 40 km north of the town of Sepon. The study area was selected from the Sepon Basin, Vilabouly District, Savannakhet Province, Lao PDR. The Sepon Basin is well known for the copper–gold mineralization district, as shown in Figure 1, the location of the project, and the Cu-Au mineralization within the SMD occurred along the east-trending corridor, approximately 40 km long and 10 km wide. The open-pit gold and copper mining activities and productions in the SMD were commenced in late 2002 and late 2004, respectively, within indicated and inferred resources of 83 Mt @ 1.8 g/t Au for 4.75 Moz plus 100 Mt @ 2% Cu for 2.0 Mt [26,27].

### 2.2. Geology and the Structural Geology

Geological assessment of the Indochina Plate throughout geological time has played a crucial role in creating a favorable environment for forming copper–gold deposits in the Sepon region. The mineralization process in this area is controlled by a normal fault that exhibits a steep inclination and vertical dip. This fault configuration provides ideal conditions for hydrothermal fluids to migrate along the fault plane and permeate into the surrounding rocks. Additionally, strike-slip faults that traverse the region have contributed

to the formation of Cu and Au mineralization, along with associated minerals such as malachite, azurite, and chalcocite. Figure 2 shows the schematic model of mineralization styles in the Sepon Mineral District (after Sillitoe 1990).

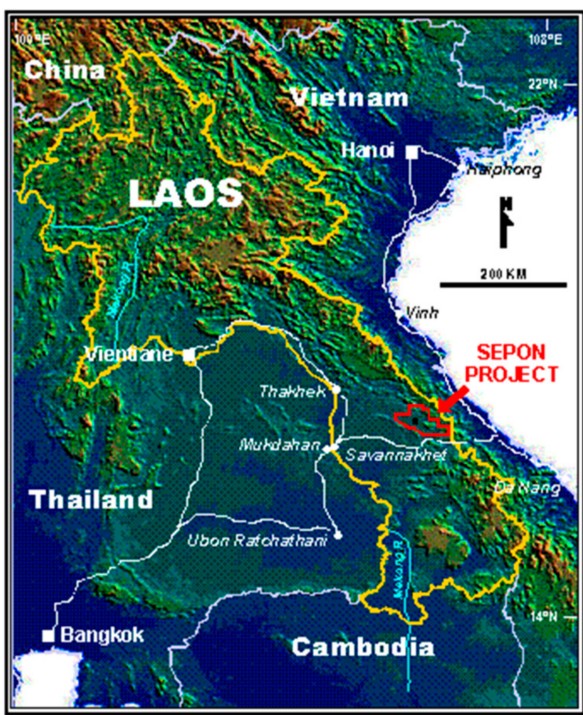

**Figure 1.** Location of Sepon Project.

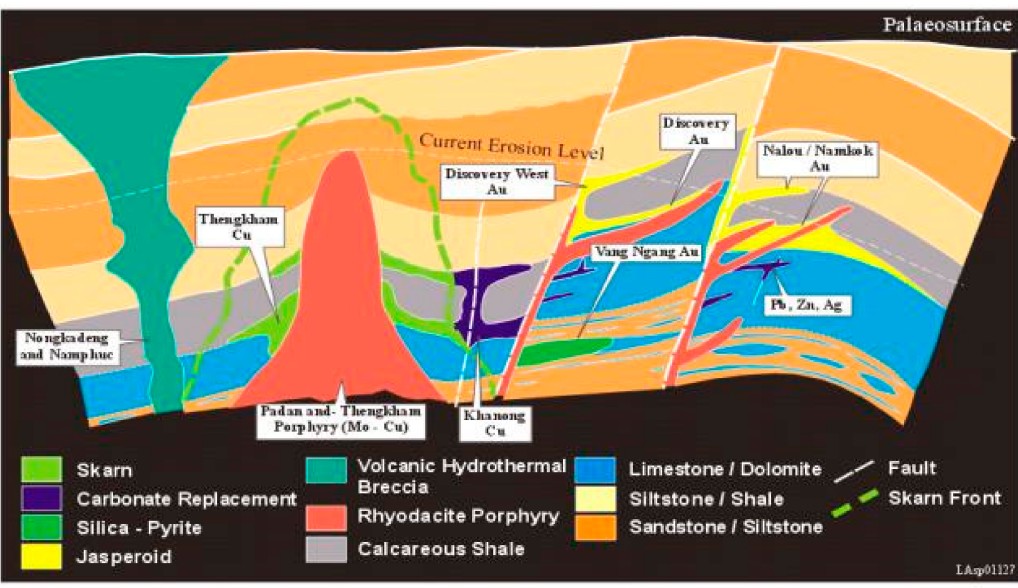

**Figure 2.** Schematic model of mineralization styles in the Sepon Mineral District [28].

The in situ supergene enrichment of sulfide minerals, particularly chalcocite, has resulted in the formation of a high-grade chalcocite enrichment blanket. Directly beneath this blanket is a zone rich in high-grade copper oxides, including minerals such as malachite, azurite, cuprite, and small amounts of native copper [28].

## 2.3. Alteration and Mineralization

Three major types of mineralization have been recognized in the SMD. Therefore, the characteristics of wall rock alteration and mineralization are different in each environment and hydrothermal process.

1.  Center on RDP intrusion: Cromie [29] has classified four types of alteration in the RDP intrusion centers (i.e., Phadan and Thenkham); (a) K-feldspar dominantly replaced the primary (igneous) plagioclase in the phenocryst and groundmass; (b) white mica with minor chlorite overprinted K-feldspar alteration; (c) quartz vein with K-feldspar alteration halo and later by quartz–chlorite–pyrite–molybdenite–chalcopyrite–bornite–hematite vein; and (d) massive barren quartz vein, vein stockwork. Cu and Mo sulfides are mainly present in the vein quartz and disseminated together with K-feldspar and white mica-chlorite alteration assemblages. However, the grades of Mo and Cu are slightly low, ranging from 40 to 1020 ppm Mo and <0.1 to 0.5% Cu, identifying sub-economic Mo and Cu mineralization in the SMD [30].

2.  Contact between wall rock lithologies and RDP: the contact between wall rock lithologies, particularly carbonated rocks (e.g., calcareous shale, limestone, and dolomitic limestone) of the Nalou and Kengkuek Formations, as shown in Figure 2, and RDP intrusions resulted in Cu-(Au) skarn mineralization. Cannell, Smith, Cromie, and Seedorff [27,29,31] classified three types of skarn mineralization and alteration assemblage: (a) garnet with minor pyroxene, biotite, and K-feldspar assemblages of prograde skarn, which were overprinted by (b) chlorite, epidote, hematite, calcite, and sulfide (e.g., chalcopyrite–bornite–molybdenite ± Au); and (c) quartz, calcite, fluorite, and pyrite ± Au in the later stage. Primary copper mineralization presented in semi-massive sulfide zones is typically less than 1% Cu, though locally, up to 5% Cu is present in chalcopyrite-rich zones [27]. Lower-grade gold, up to 1 ppm, was identified as a solid solution with chalcopyrite, and higher-grade gold, up to 290 ppm, occurred as a solid solution in pyrite crystal structure during the later stage.

3.  Sediment-hosted gold deposit: distal distance from the RDP intrusion centers, where the sets of dike and sill intruded calcareous mudstone and calcareous shale of the Discovery Formation presented sediment-hosted gold deposits as shown in Figure 2. Steep faults and secondary shallow to moderately dipping fault structures enhanced the RDP intrusions and elevated hydrothermal fluid ascending, resulting in intensely silicified carbonated rocks or jasperoid and decalcified shale. Gold is predominantly found as invisible gold or gold in solid solution with the crystallization of pyrite, most often disseminated and fracture controlled together with jasperoid and decalcified shale.

## 2.4. Geotechnical Rock Mass Characterization

Before commencing mining operations, it is crucial to have a comprehensive understanding of the geological strength of the rock mass. In this study, the rock mass was evaluated using the Geological Strength Index (GSI), owing to its extensive applicability in underground mining research. The GSI system, developed by Hoek in 1994, is a widely employed method for the effective, efficient, and simple characterization of rock masses. This approach was established by combining observations of rock mass conditions (relying on Terzaghi's descriptions) with relationships derived from extensive experience gained through the utilization of the Rock Mass Rating (RMR) system. The average GIS value can be determined by establishing a connection between the structure of the rock mass and the circumstances surrounding rock discontinuities. Based on the core log samples, it can be observed that the Rock Quality Designation (RQD%) demonstrates a relatively good average ranging from 80% to 100% at depths exceeding 100 to 500 m. As part of the assessment process, fieldwork and geotechnical logging were conducted on the core, as shown in Figure 3a,b.

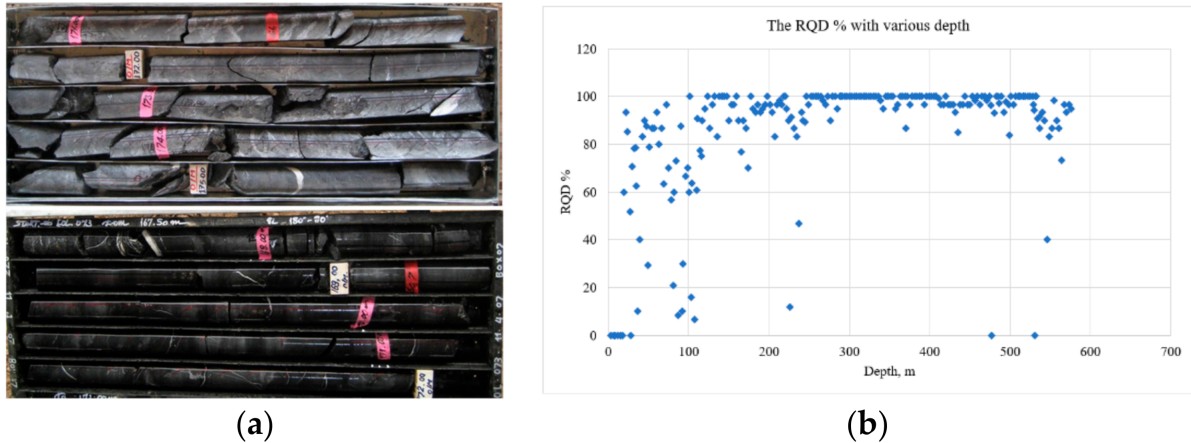

(**a**) (**b**)

**Figure 3.** Evaluation of the geotechnical characteristics of the rock mass in Sepon Mine. (**a**) Core log sample box for evaluation. (**b**) Evaluated RQD in the research area of RQD.

Laboratory studies were performed to evaluate the quality of intact rock. Among the initial strength tests conducted, Triaxial Compressive Strength Test was conducted. This involved using three sets of specimens obtained from a generally undisturbed rock sample for the pre-test and post-test illustrated in Figure 4a–c, which illustrates the pre-test and post-test specimens, respectively. The results of the triaxial tests are shown in Figure 5a–c. The findings indicate that the ore sample exhibited the highest uniaxial compressive strength (UCS), with a measurement of 224.92 MPa. Ryodacite porphyry (RDP), followed by a UCS of 82.58 MPa, and Nodular Shale with 66.59 MPa. The highest strength value corresponds to the core sample with the highest strength. Furthermore, the test results also include the calculation of Young's modulus, expressed in GPa. It is worth noting that the general densities of all three rock types were nearly identical.

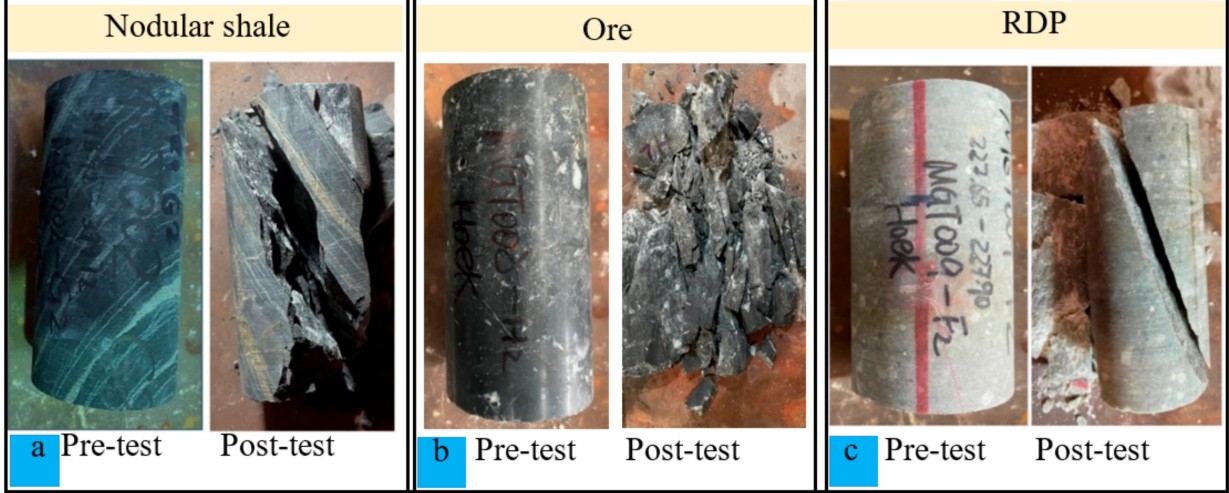

**Figure 4.** The pre-test and post-test are shown using a triaxial compressive strength test for (**a**) Nodular shale, (**b**) Ore, and (**c**) RDP.

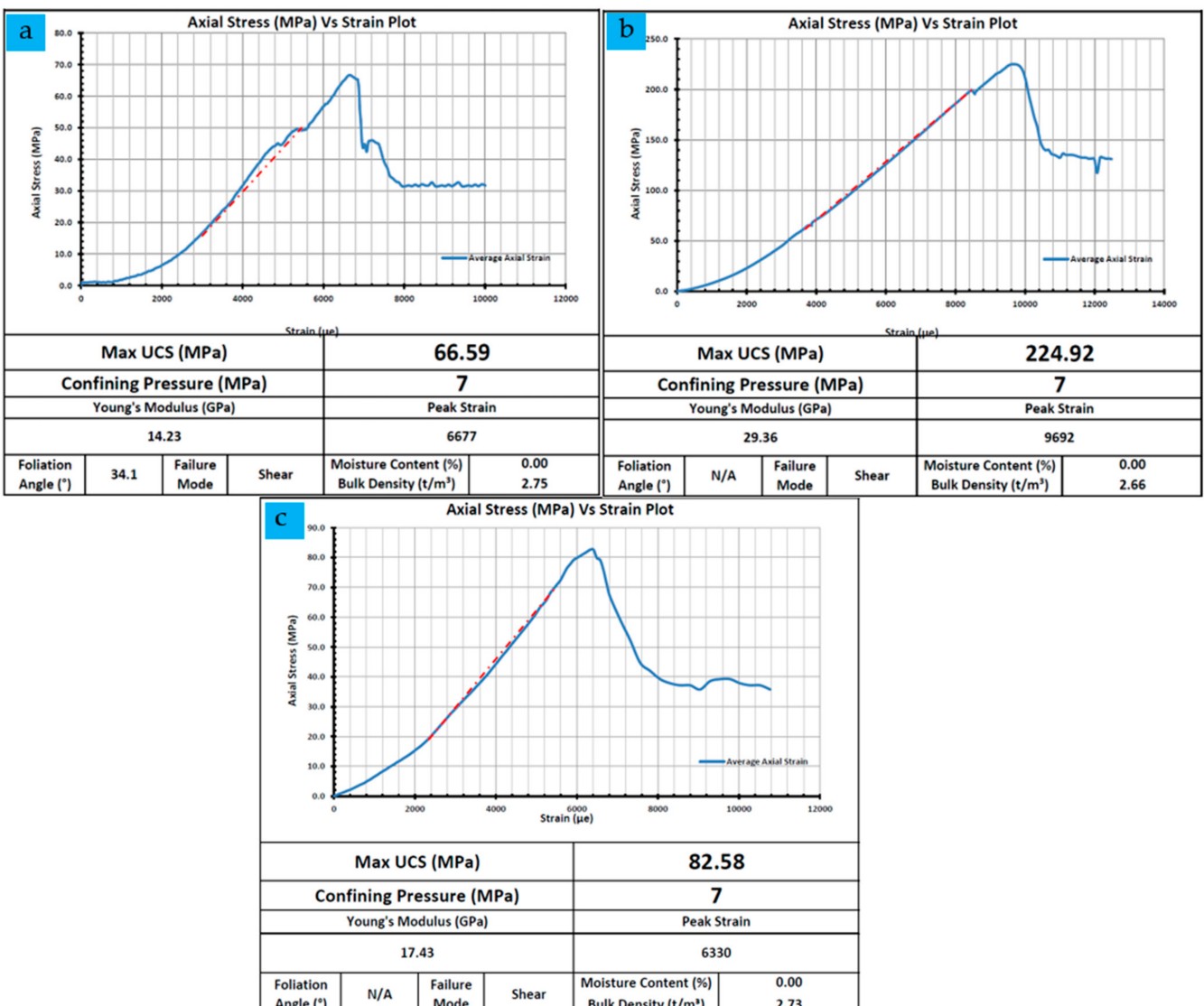

**Figure 5.** The results of the lab test for triaxial compressive strength are presented as follows: (**a**) Nodular shale, (**b**) Ore, and (**c**) RDP.

## 3. Numerical Analysis

### 3.1. The Input Parameters

The fundamental input parameters were established using the rock lab application, as indicated in Table 1. This was conducted to determine the value of the entire rock, or the rock with the core specimen as a fresh rock without weathering. The parameter for the Geological Strength Index provided in [32,33] is used. Various Geological Strength Index (GSI) values were calculated between 35 and 55, which indicate blocky rock masses with fair joint surface quality and extremely blocky rock masses with good joint surface quality.

### 3.2. Model Construction

A finite element plastic numerical analysis utilizing the Generalized Hoek–Brown criterion was performed using RS2 software to assess the stability of the surface slopes and underground stopes in a simplified model. The active underground section encompasses a former open-pit area. Figure 6 illustrates the segments of the pit bottom with an orebody dip angle of 62° connecting the footwall and hanging wall at the midpoint. The dimensions of the model were 520 m along the short axis and 700 m along the long axis. The y- and x-directions have fixed horizontal and vertical boundaries, while the upper part of the

model is left free to allow for deformation movement. In terms of design, a safety berm was retained, and the slope had a height of 20 m and a slope angle of 70°. The overall slope had an inclination of approximately 47°, and the total height was approximately 80 m.

**Table 1.** The geotechnical properties of rock at Sepon gold mine deposit.

| Geological Strength Index (GSI) | Zone/Rock Type | Unit Weight (MN/m³) | $\sigma_{ci}$ (MPa) | Tensile Strength (MPa) | $E_{rm}$ (GPa) | v | Hoek Brown Parameter | | |
| --- | --- | --- | --- | --- | --- | --- | --- | --- | --- |
| | | | | | | | $m_b$ | s | a |
| 72 | Footwall RDP | 0.0275 | 82.58 | 15.59 | 17.43 | 0.3 | 6.167 | 0.0315 | 0.501 |
| 62 | Ore body | 0.0266 | 224.92 | 13.81 | 29.36 | 0.3 | 2.656 | 0.010 | 0.502 |
| 51 | Hanging wall Nodular shale | 0.0279 | 66.59 | 11.66 | 14.23 | 0.35 | 0.898 | 0.001 | 0.505 |

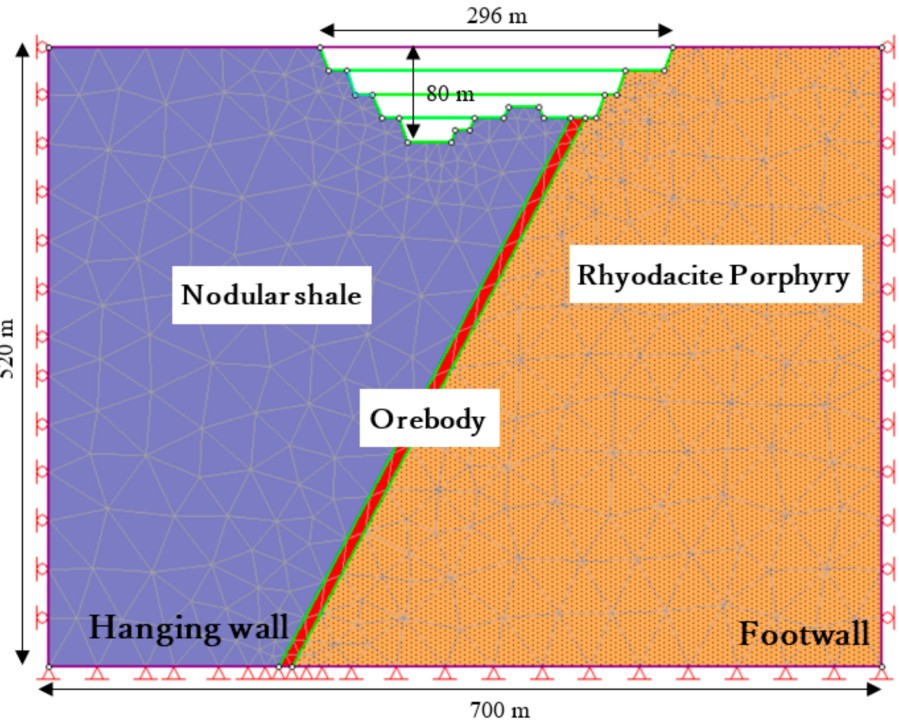

**Figure 6.** Simplified model set-up showing dimensions.

Specific measures are taken in the design to prevent surface subsidence, such as incorporating a crown pillar thickness of approximately 50 meters [34,35]. The stope was 5 m wide and 5 m high. The mining sequence involved leaving sill pillars after five excavations at each level to provide support for the working floor, followed by backfilling. In certain areas of stopes, additional support through shotcrete and rock bolting, along with continuous monitoring, is necessary for stability management. With a GSI of 62 and a vein dip of 62°, mining operations utilize the overhand-cut method. Unlike conventional mining sequences that progress from the bottom to the top, leaving a 5-meter-thick sill pillar between each excavation, the overhand cut method starts from the bottom and moves upward to the top. This process is depicted in Figure 7, with the first group of excavations comprising five steps. Once the initial group of excavations was complete, the method was repeated for subsequent excavations using the overhand-cut approach. This mining sequence continued until the final excavation was completed.

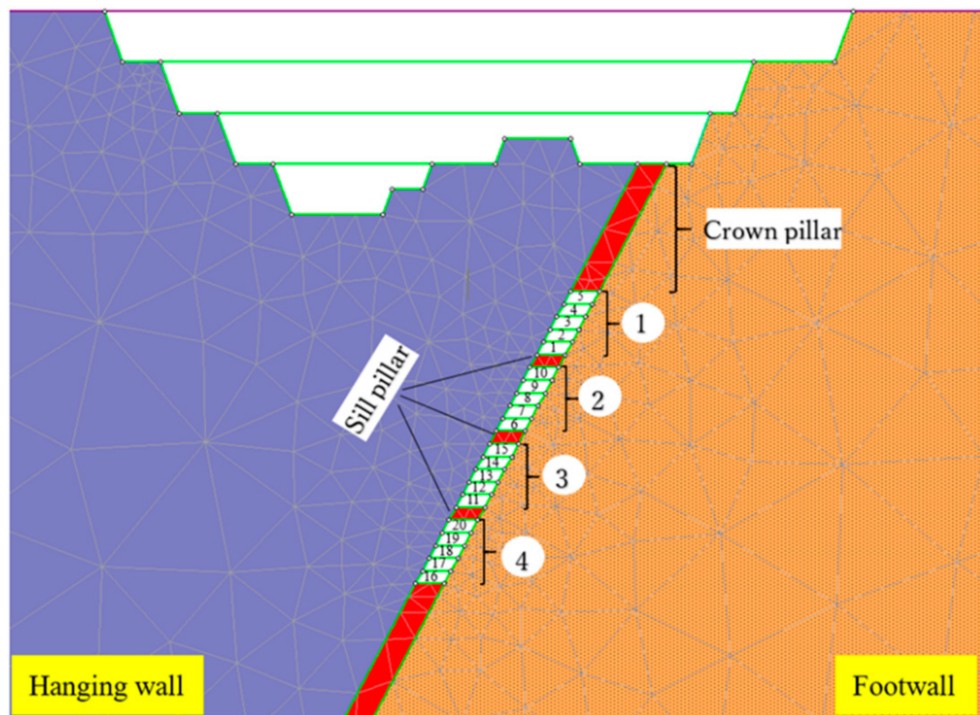

**Figure 7.** The mining consequence.

## 4. Results and Discussion

In terms of rock mechanics and rock engineering, the displacement and yielded elements were tracked in this study based on a numerical model to assess stability with criteria based on plastic strain. It specifies that a 2.5% plastic strain is an indicator of failure in a non-entry stope [36]. As a result, the goal of this study was to examine how a designed stope might affect the slope face and pit bottom. The analysis results in a two-dimensional numerical model showed that stresses and stope deformation occurred more on the side of the hanging wall than on the footwall.

### 4.1. Surface and Slope Deformations

The stability of the slopes was examined using monitoring locations near the pit surface, which resulted in a factor of safety (FoS) of approximately 2.46, as shown in Figure 8. Underground mining operations must be continuously monitored once the surface slopes have finished. According to the results in Figure 9, a high displacement mostly occurred in the hanging wall. The displacement monitoring pins placed every ten meters along the surface are shown. The display revealed that the footwall was unaffected, but the bench's high displacement on the hanging wall side dramatically grew to the pit bottom.

### 4.2. Comparison of Different Vein Widths

#### 4.2.1. Yielded Elements

Six different design factors are carried out in different sizes as open stope dimensions to assess and choose the best design based on geological conditions and ore recovery. The elements produced were used by numerous researchers to assess the likely failure zone [37–39]. Therefore, this study was conducted in the yield zone to evaluate the potential failure zones.

According to the results, the failure zone is in a tiny location, as indicated in the yielded elements. The excavation's first step is 3 × 15 m and comprises five minor steps. Figure 10a shows the small-scale propagation surrounding an open stope with a size of 3 × 15 m (wide and high). Each tiny step in the mining sequence was 3 × 3 m in size, using

an overhand cut that was adequate for the rock mass and considered the orebody's GIS orebody value of 62. The ore recovery is reduced, and the first design is rather narrow, but it exhibits great stability around the open stope and has no impact on the crown pillar. The results in Figure 10b also demonstrate small-scale spreading around the open stops, but the size of the stope was slightly different, being $3 \times 25$ m for the initial step of excavation, which is made up of five smaller steps, each of which is $3 \times 5$ m high. In this situation, the failure zone increases more than the result in Figure 10a for the initial design. However, even though the stability was slightly higher than before, the crown pillar was unaffected. However, the ore recovery has improved. The design shown in Figure 10c utilized an excavation of $5 \times 25$ m in size as its initial phase. According to the results, the failure zone primarily formed from the first excavation to the fourth excavation, more so than in the previous two designs. As shown in Figure 10b, the failure zone increased twice as much as that of the design. Additionally, there was a propensity for failure on the higher slope due to open stope failure, but there was no response along the crown pillar. However, the ore recovery has improved. Figure 10d–f almost all has the same failure zone, which starts to grow from Figure 10e to Figure 10f. According to the findings, it is evident that large open stopes can result in significant failure zones, particularly around the hanging wall and crown pillar failures at designs as large as $20 \times 25$ m.

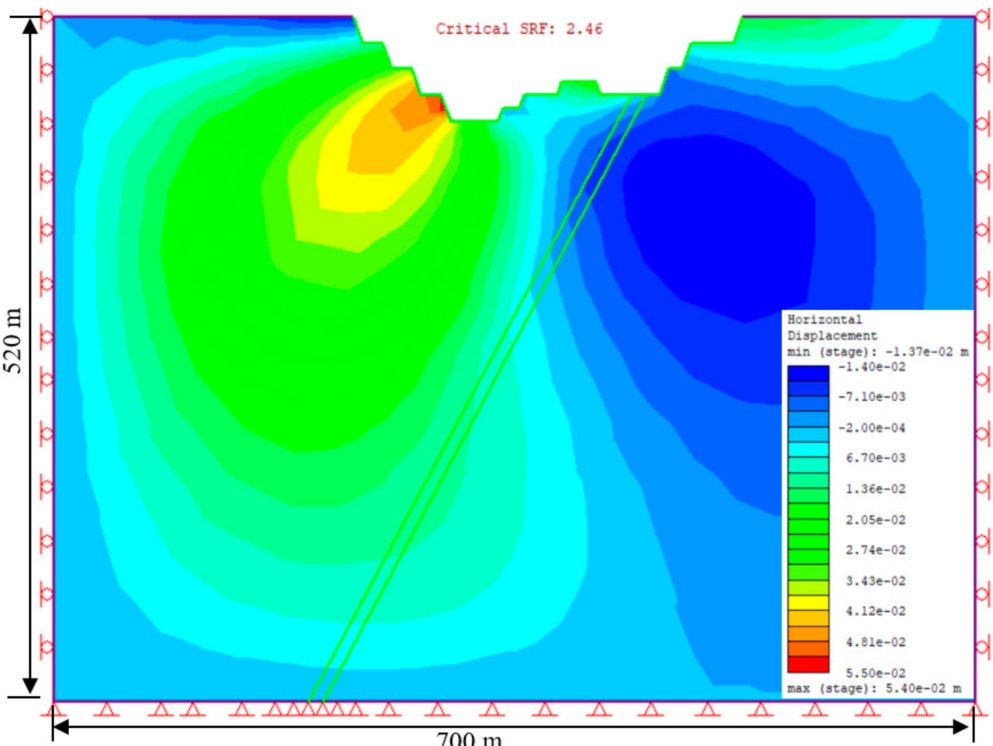

**Figure 8.** Slope stability before beginning an underground mining operation.

Based on the comparison of six different designs, considering geometry and geological features, it can be concluded that Figure 10c represents a suitable stope design for the Sepon mine. The results indicate that the failure zone in this design is within an acceptable range, neither excessively high nor unacceptably low. This finding suggests that Figure 10c provides a balanced and feasible stope design for the specific conditions at the Sepon mine.

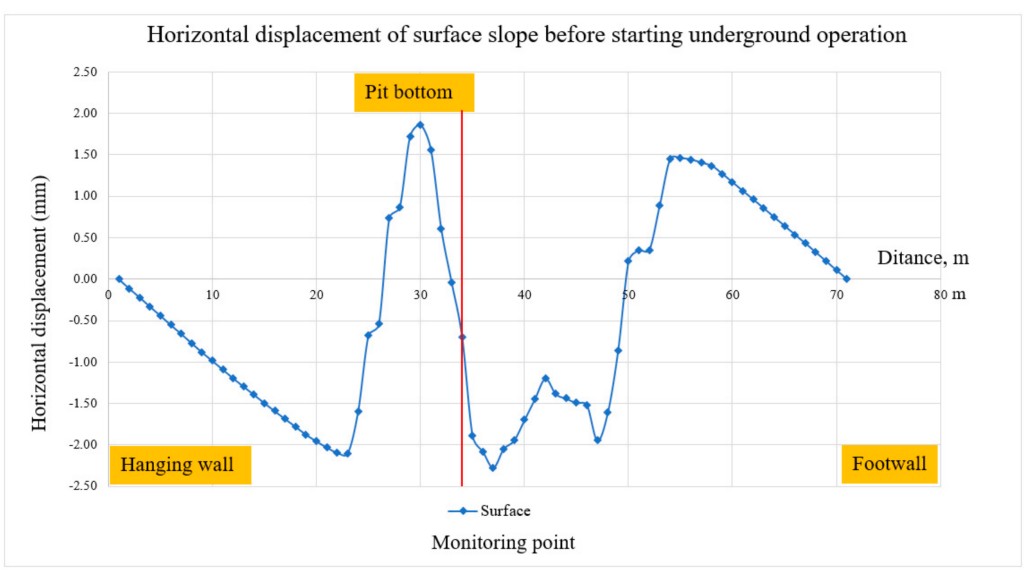

**Figure 9.** The monitoring point along the surface displays displacement.

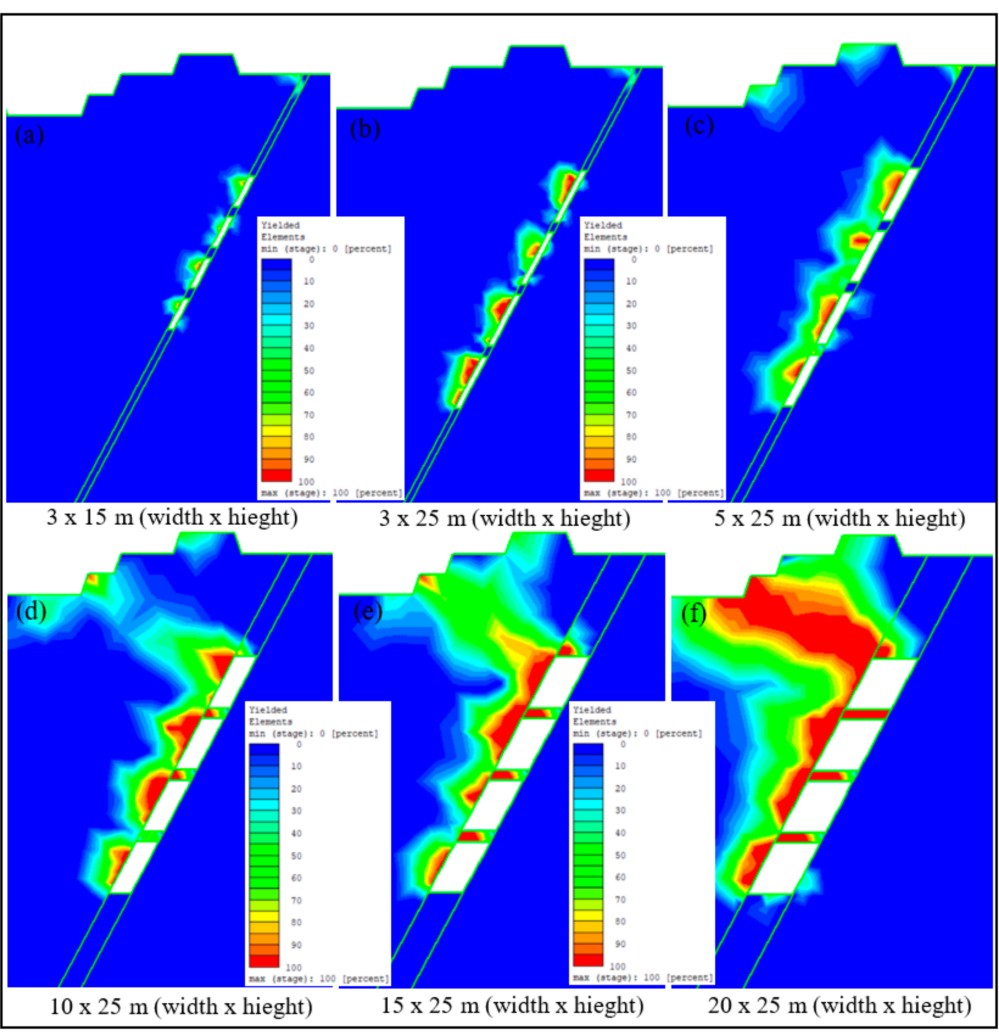

**Figure 10.** Various designs of open stop dimension.

4.2.2. Displacement

Mining operations induce volumetric and stress–strain changes in rock masses. As the deformation exceeds the limits set by the rock's strength, instability develops around the excavation area. It is crucial to monitor the stability of the surrounding rocks to prevent unforeseen failures. Therefore, careful observation and monitoring of the stability of rocks encircling stopes are essential. In contrast to real-time monitoring during active mining operations, where automatic monitoring occurs intermittently and occasionally using reference points, in this simulation, the monitoring points were strategically placed at critical locations in the model. Evaluating displacements in the ongoing underground Sepon Mine revealed significant deformation conditions that could lead to failure. Based on these results, it can be observed that the displacements along the surface were pinned every 10 m, as shown in Figure 11. The highest displacement occurred on the hanging wall, predominantly on the crest, whereas the footwall side exhibited relatively lower displacements. This can be attributed to the stronger rock mass on the footwall side and the weaker rock mass on the hanging wall. This study examined six different designs for an open stope. The wider open stope design (20 m × 25 m) resulted in a larger failure zone and higher displacement. Conversely, a stope with 3 × 15 m dimensions exhibited the lowest potential failure zone. The results indicated that the displacement gradually increased with increasing stope size.

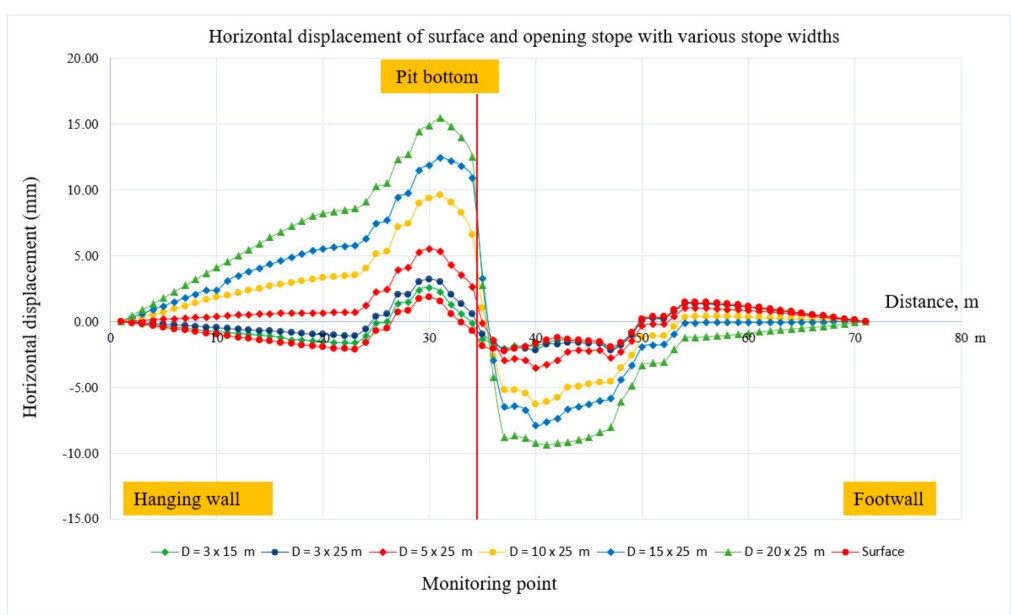

**Figure 11.** Comparison of the displacement of six different stope sizes.

Figure 12 illustrates the displacement, indicating that the highest displacement occurred during the third and fourth excavation steps, particularly on the hanging wall. However, on the footwall, the displacement remained generally low. Additionally, the safety factor remained relatively stable throughout the excavation process, with an approximate value of 1.67. According to the results, the first excavation step showed the lowest displacement, with a value of 2.37 mm. The second open-stope excavation exhibited a gradual increase in displacement to 2.63 mm. Similarly, the third and fourth steps of the excavation showed further increases in displacement, measuring 3.25 mm and 3.78 mm, respectively. It can be observed that as the excavation depth increases, the predominant displacement also increases [37,38]. Furthermore, the highest displacement occurred on the hanging wall side, whereas the footwall side displayed a slightly higher displacement, especially on the crest, as shown in Figure 13.

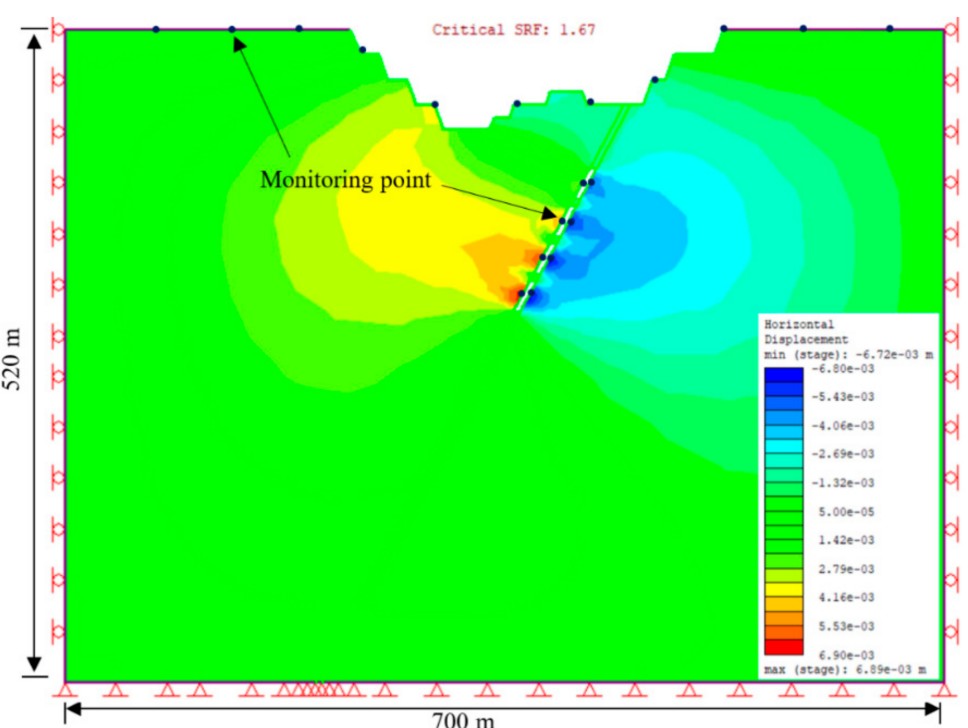

**Figure 12.** The displaying displacement in the open stope.

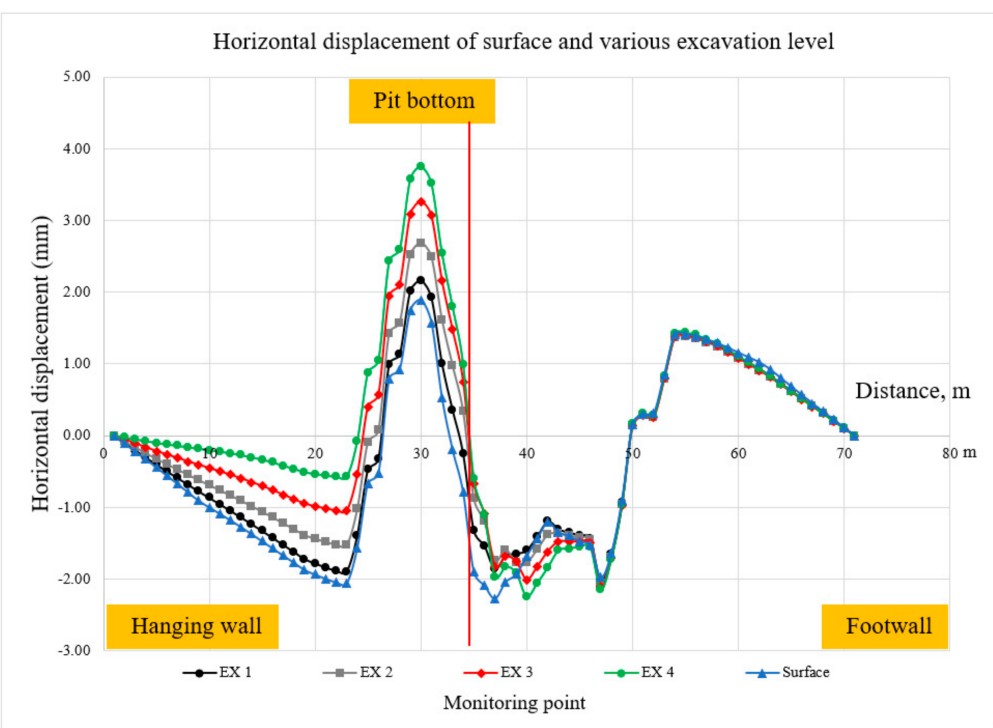

**Figure 13.** Horizontal displacement of surface and various excavation levels.

Based on the data displayed in Figure 14, the monitoring sites were positioned along the side of the open stope. High displacements were generally observed on the hanging wall side, which can be attributed to the lower strength of the rock mass compared to the footwall side. Specifically, the hanging wall side exhibited a Geological Strength Index (GSI) of 51, indicating a relatively weaker rock mass. In contrast, the footwall side had a

GSI of 72, indicating a stronger rock mass. Consequently, this and the favorable geometry recorded low displacements on the footwall side in this scenario.

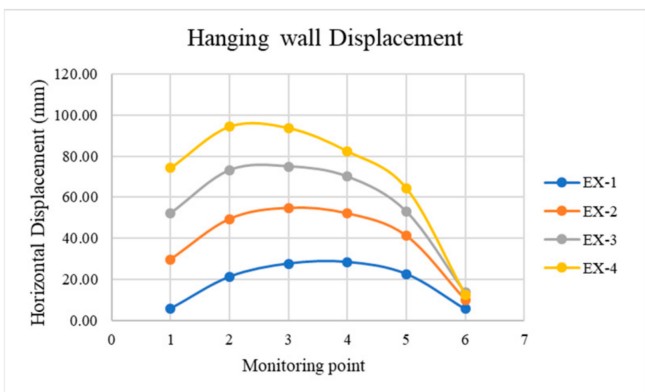 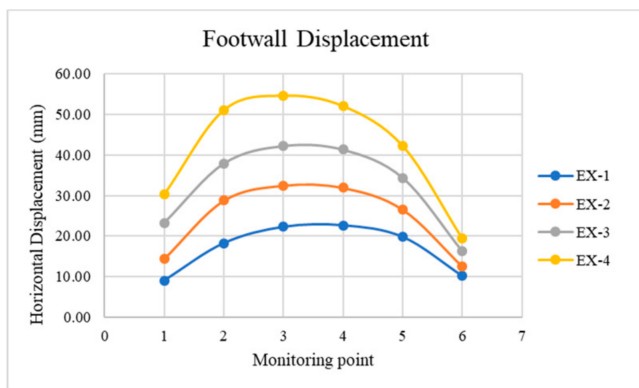

**Figure 14.** The horizontal displacement in each stope excavation.

*4.3. Determined of the Various Stress Ratio*

According to the results, it can be observed that the instability around the open stope is relatively low at a K ratio of 0.5 compared to other stress ratios, as shown in Figure 15a. In Figure 15b, it can be observed that the predominant failure starts developing around the open stope with a K ratio of 1. The failure zone was more pronounced on the hanging wall side because of the weak rock mass with a GSI of 51. When the K ratio increased to 1.5, more failure zones began to develop, as shown in Figure 15c. Furthermore, as shown in Figure 15d–f, the instability dramatically increased when the K ratio was increased from 2 to 3. The most significant failure zone occurred at the highest K ratio of 3, surpassing the failure zones observed at K ratios of 2 and 2.5. However, the sill pillar remained stable for K ratios of 0.5 and 1. At a K ratio of 1.5, the sill pillar was affected, along with the crown pillar. Additionally, for K ratios of 2 to 3, both the sill pillar and crown pillar are significantly affected. Based on the result of the horizontal displacement graph in Figure 16, it can be said that the low displacement depends on the stress ratio. In this study, a K ratio of 0.5 with a displacement of 8.32 mm was followed by a K ratio of 1, which had a double increase of 16.7 mm. While with a K ratio of 1.5, the result in displacement was 35.4 mm, it almost doubled to 62.8 mm with a K ratio of 2. However, when the K ratio increased to 2.5, the displacement slightly increased to 87.9 mm. In addition, it increased dramatically to 135 mm with a K ratio of 3. Consequently, the highest displacement occurred with the highest stress ratio [37,40,41]. However, the lowest displacement occurred along with the lowest stress ratio.

*4.4. Different GSI Values*

According to the outcome depicted in Figure 17, the geological strength index (GIS) affects the safety factor. As can be seen from the displays, underground mining that complies with a GSI of 35 is considered safe when a factor of 1.3 is used. However, when the GSI is lower than 35, the safety factor falls below 1.3, which fails to meet the requirements. However, it is evident from the horizontal displacements shown in Figure 18 that the lowest GSI value resulted in up to 15 cm displacements, as illustrated by the highest displacement at GSI 30 and below. However, as the GIS values increase, higher GSI values correspond to smaller horizontal displacements [37]. This confirms that a high GSI value leads to a high level of stability, particularly in surface mining and open stopes. Thus, the GSI value plays an important role in the numerical simulation of the safety of the factors and horizontal displacements.

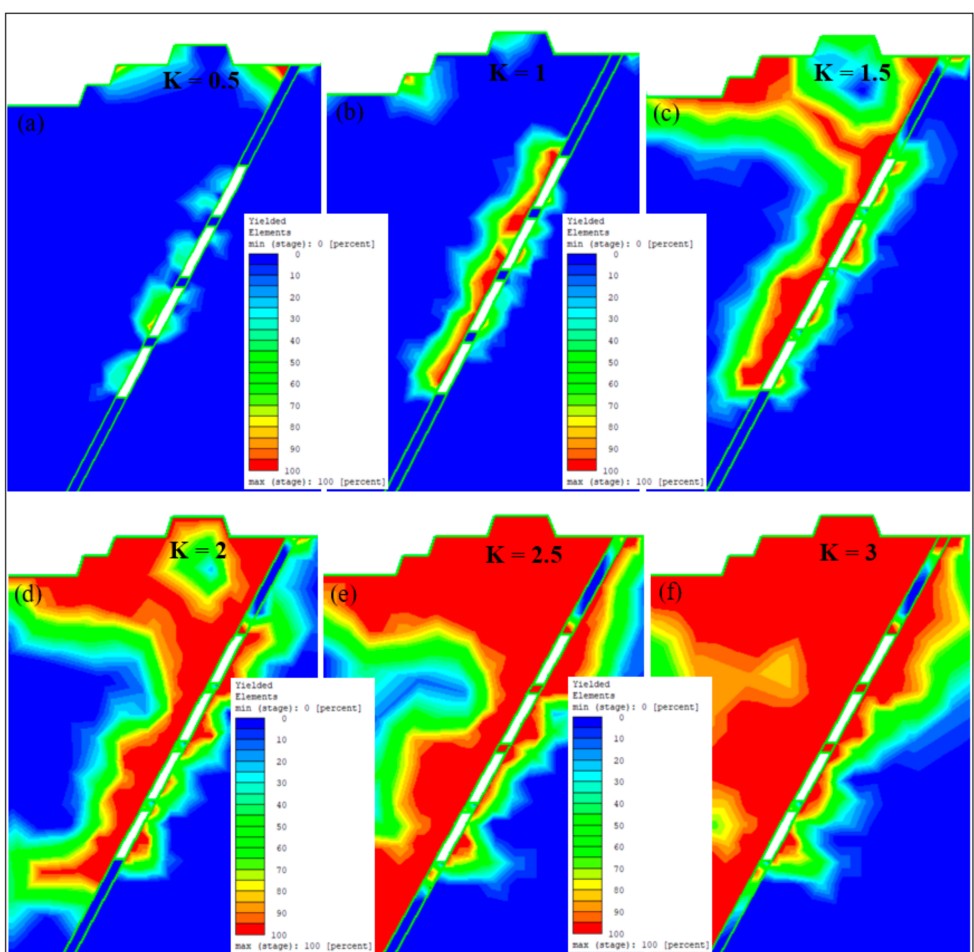

**Figure 15.** Depict the result of the study area under different stress ratios under yielded element.

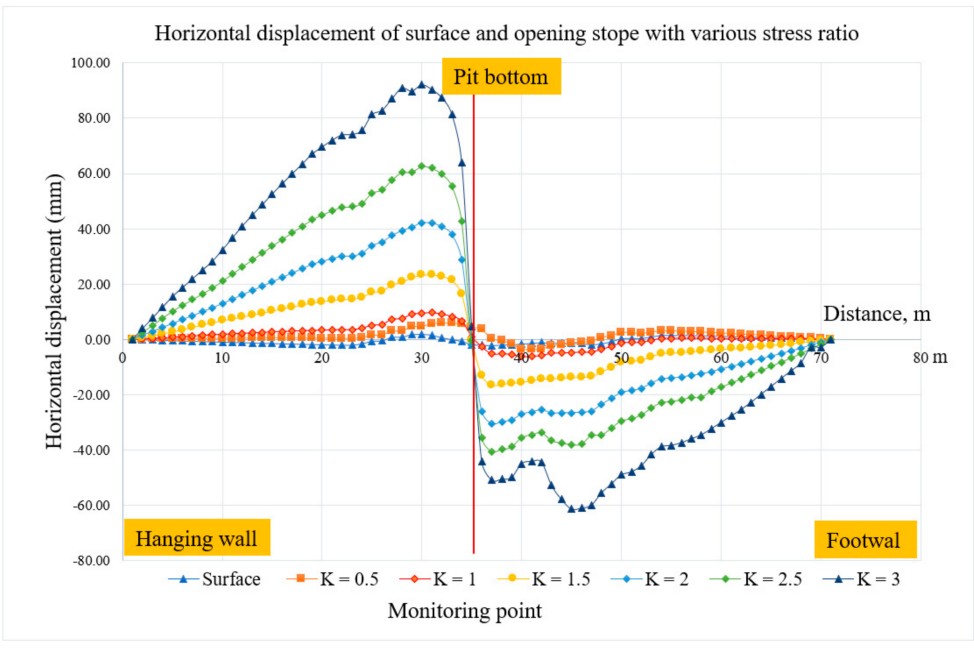

**Figure 16.** The horizontal displacement with various stress ratios.

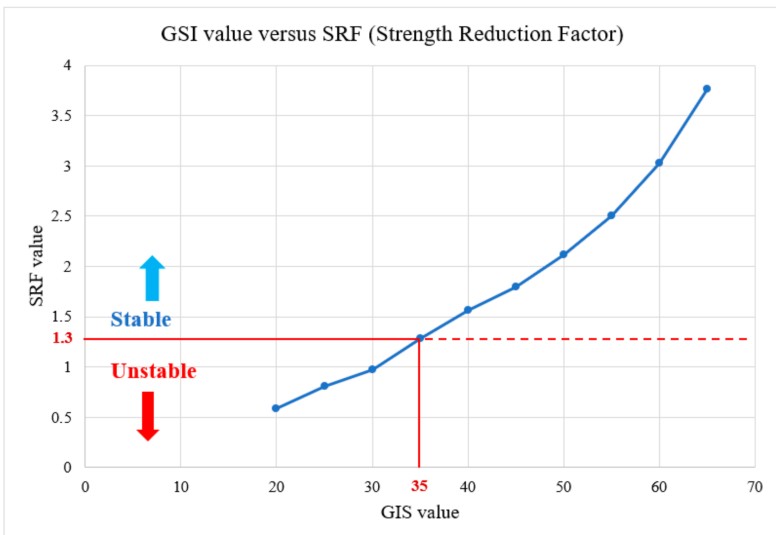

**Figure 17.** Strength reduction factor with various GSI values.

The results obtained from the simulation presented in Figure 19 provide valuable insights into the behavior of the open stope under different GSI conditions. In Figure 19a, the lowest GSI of 25 leads to the occurrence of a considerable failure zone, primarily affecting the hanging wall side, with a relatively minor impact on the crown pillar. This suggests that the stability of the open stope was significantly compromised at this GSI level. As shown in Figure 19b, an increase in the GSI to 30 resulted in a slight improvement in the failure zone. Although there was a marginal reduction in the extent of failure, the hanging wall region continued to exhibit dominant failure behavior. Notably, even a moderate increase in GSI had a limited effect on stabilizing the open stope. However, the scenario significantly changed as GSI continued to increase. From the GSI values of 35 to 50, there was a noteworthy decrease in the failure zone size and extent. Notably, Figure 19e,f shows that when the GSI reached 45 and 50, respectively, the impact on the failure zone was the least pronounced. These figures indicate that the open stope exhibited improved stability with a considerable reduction in the occurrence of failures, particularly in the hanging wall portion.

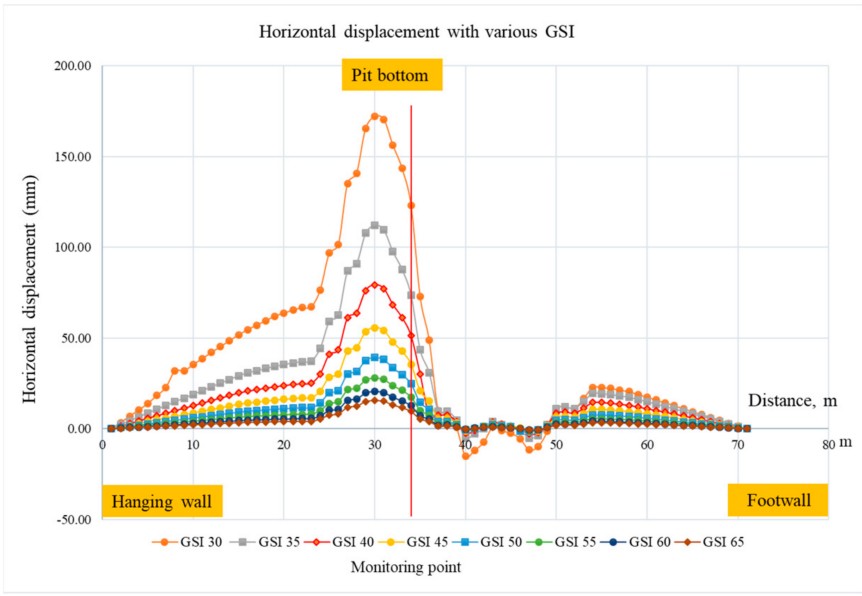

**Figure 18.** Horizontal displacement with various GSI values.

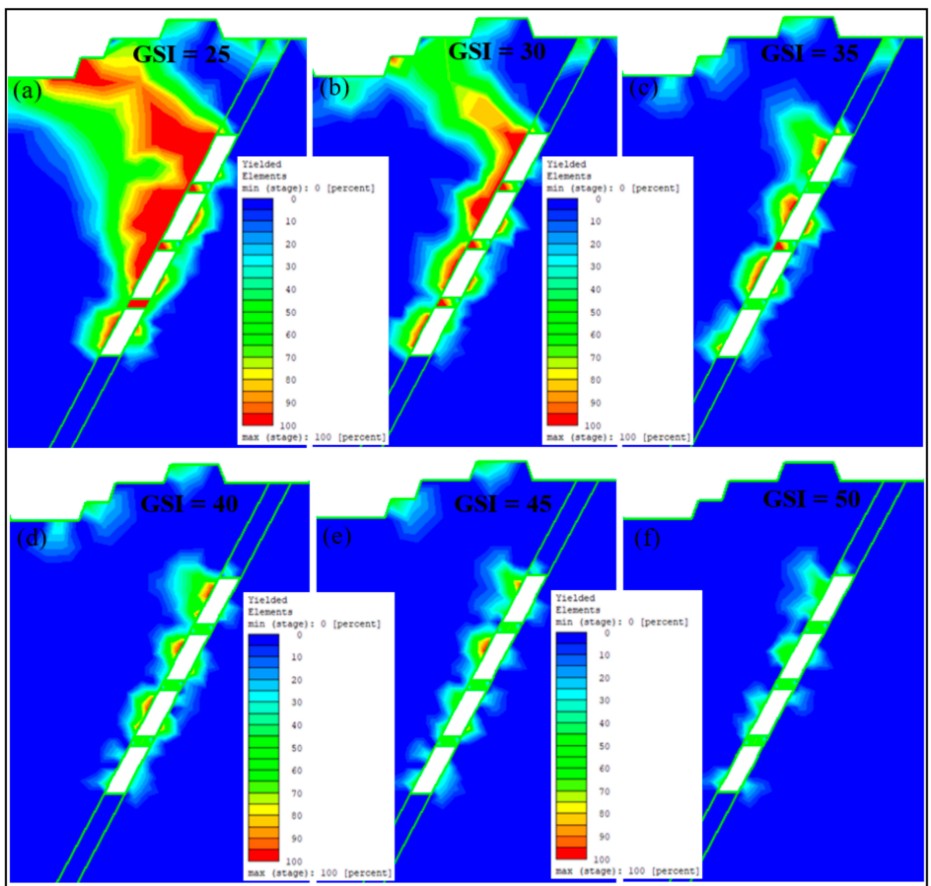

**Figure 19.** The effect of various GSI values to open stope.

Overall, the simulation results presented in Figure 19 highlight the critical role of GSI in determining the stability of the open stope. A lower GSI, such as 25, led to a significant failure zone, primarily affecting the hanging wall side. In contrast, higher GSI values, such as 45 and 50, resulted in improved stability and a substantial decrease in the failure zone. These findings underscore the importance of considering the GSI as a crucial factor in designing and managing open stopes to ensure their long-term stability and safety.

## 5. Conclusions

This study was conducted to assess the impact of an open-stope excavation beneath an existing open pit. The objective was to understand the patterns of slope deformation and the influence of the open stope beneath the open pit bottom, which are crucial mining characteristics. Throughout the investigation, a 2D isotropic plastic numerical model using RS2 was created to represent both the underground and open pit sections of the mine. Based on the results and discussion, several significant issues can be concluded:

- Surface and Slope Deformations: The stability of the slopes was thoroughly examined, revealing a factor of safety (FoS) of approximately 2.46. Displacement monitoring revealed that significant displacement primarily occurred in the hanging wall, while the footwall remained relatively unaffected. This highlights the importance of assessing and managing slope stability, particularly in relation to the hanging wall.
- In the comparison of six different vein widths, the study evaluated six designs for open stope dimensions. The findings revealed that wider open stopes were associated with larger failure zones and higher displacements. On the other hand, narrower designs showed lower potential failure zones, although displacement gradually increased with larger stope sizes. Based on the analysis, it was determined that a stope dimension of $5 \times 25$ m was considered appropriate. This design struck a balance between avoiding

excessive failure risks associated with larger dimensions and ensuring sufficient stability. By selecting this stope dimension, this study aimed to mitigate the potential for significant failures while also taking operational considerations into account.

- Determination of Various Stress Ratios: The impact of different stress ratios on instability around the open stope was analyzed. Lower stress ratios showed relatively low instability, while higher stress ratios above 1.5 led to significant failure zones. The hanging wall side was found to be more prone to failure due to a weaker rock mass. This highlights the need to consider stress ratios and their influence on stability when designing and managing open stopes.
- Different Geological Strength Index (GSI) Values: The geological strength index (GSI) was found to have a notable influence on safety factors and horizontal displacements. Lower GSI values resulted in larger failure zones and higher displacements, while higher GSI values (above 35) improved stability and reduced failure zones [37], particularly in the hanging wall. Understanding the GSI of the rock mass is crucial for assessing stability and implementing appropriate design strategies.
- To ensure stability, it is recommended that the crown pillar thickness not be less than 40–50 m and the sill pillar have a minimum thickness of 5 m. These recommendations are based on the geological conditions specific to the mine and are essential for maintaining the stability of the underground workings.

These conclusions emphasize the importance of considering slope stability, stope design dimensions, stress ratios, and GSI values when assessing and managing the stability of open stopes in underground mining operations. By understanding the behavior and potential failure mechanisms associated with these factors, effective design strategies can be implemented to ensure the long-term stability and safety of mining operations. Additionally, a comprehensive analysis that examines both underground and open-pit sections is necessary to gain a better understanding of the impact of underground mining on overall mine stability.

**Author Contributions:** Conceptualization, S.P. and H.S.; data curation, S.P. and A.H.; formal analysis, S.P.; methodology, S.P. and H.S.; software RS2, S.P. and A.H.; supervision, H.S. and T.S.; validation, H.S., T.S. and A.H.; visualization, S.P.; writing—original draft preparation, S.P.; writing—review and editing, H.S., T.S., A.H., P.P., S.S. and K.S.; materials, H.S., T.S. and A.H. All authors have read and agreed to the published version of the manuscript.

**Funding:** This research work is part of a doctoral program supported by JICA.

**Data Availability Statement:** Research data will be provided upon request from the first author.

**Acknowledgments:** We sincerely thank JICA for the scholarship funding that has facilitated this research. Laboratory for providing the RS2 software used for analysis and test results of rock mechanics, some data, and general information from the Spon Company.

**Conflicts of Interest:** The authors declare no conflict of interest.

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
