# Peer review of "A Stope Mining Design with Consideration of Hanging Wall When Transitioning from Open Pit Mining to Underground Mining for Sepon Gold Mine Deposit, Laos"

_mining, doi:10.3390/mining3030027_

Round 1
Reviewer 1 Report
Paper no: mining-2507356
Title: A stope mining design with consideration of hanging wall when transitioning from open pit mining to underground mining for Sepon gold mine deposit, Laos
In the paper the analysis of the transition of mining activities from the surface to underground at the Sepon Gold mine in Laos is studied. The idea seems interesting, but the novelty is doubtful. What new is in the approach presented in the study? The novelty should be clearly defined in the introduction part of the manuscript. Moreover, several aspects should be corrected in the in-depth revision of the manuscript:
· The schematic model of mineralisation styles in the Sepon Mineral District presented in Fig.2 should described in detail in the revised version of the manuscript.
· Fig. 5 is illegible. Please correct or remove in the revised version of the manuscript.
· SI units should be used in the whole manuscript.
The conclusions should reflect the research presented in the R&D part. Please reformulate in the revised version of the manuscript.
Author Response
Dear reviewer 1,
Thank you very much for your careful and professional suggestions. Your comments have been extremely valuable and helpful in revising and improving our paper. We have carefully studied your comments and made the necessary corrections, hoping they meet with your approval. We have also provided a point-by-point response to address each of your comments as following:
Comment 1: In the paper the analysis of the transition of mining activities from the surface to underground at the Sepon Gold mine in Laos is studied. The idea seems interesting, but the novelty is doubtful. What new is in the approach presented in the study? The novelty should be clearly defined in the introduction part of the manuscript. Moreover, several aspects should be corrected in the in-depth revision of the manuscript:
Response 1: We apologize for any confusion. I have carefully studied your response. Although defining the novelty of your paper can still be challenging, I appreciate your effort in extending the introduction section as highlighted in yellow. This extension aims to emphasize the novelty of your research after conducting a thorough literature review. In this study, we examine surface stability assessment in open pit mining operations, considering subsequent underground activities. Our research provides a comprehensive approach to ensure long-term stability and safety. The unique contribution of our study lies in the emphasis on continuous monitoring during underground operations. By addressing surface stability before starting underground mining and implementing ongoing monitoring, we fill a crucial gap in the literature, contributing valuable insights for more effective stability strategies.
Comment 2: The schematic model of mineralisation styles in the Sepon Mineral District presented in Fig.2 should described in detail in the revised version of the manuscript.
Response 2: We have included additional details in the manuscript, which have been highlighted in yellow (page 4-5). These added details aim to further enhance the content and provide a clearer understanding of the research being presented.
Comment 3: Fig. 5 is illegible. Please correct or remove in the revised version of the manuscript.
Response 3: We have made edits to the manuscript and added a new figure (page 5) to enhance the content and presentation. We believe these changes contribute positively to the overall quality of the paper.
Comment 4: SI units should be used in the whole manuscript.
Response 4: We have reviewed the SI units in the manuscript and made necessary adjustments. However, there may still be inconsistencies between the units mentioned in the captured picture and those presented in Table 1 (page 7), specifically regarding the unit weight. We apologize for any discrepancies and will ensure that the final version of the manuscript maintains consistency in unit usage throughout.
Comment 5: The conclusions should reflect the research presented in the R&D part. Please reformulate in the revised version of the manuscript.
Response 5: We have reviewed the revised version of the manuscript and noted the edits made to the conclusion section. The changes have been highlighted in yellow on pages 16-17, indicating the last part of the conclusion.
Once again, please allow us to express our thanks to reviewer for your careful and professional comments, which are very helpful to improve our manuscript. All the other Suggestions have been improved accordingly in our paper. All the content modification proposed by the reviewer, we have marked it as red. We tried our best to improve the manuscript and made some changes in the manuscript. These changes will not influence the content and framework of the paper. We appreciate your warm work earnestly and hope that the correction will meet with approval.

Reviewer 2 Report
This study focuses on the transition from surface to underground mining at the Sepon Gold mine. It assesses the stability of surface slopes and designs underground mining methods based on ore body characteristics. The stability of surface slopes and opening stopes is evaluated using numerical analysis. The pit design ensures stability with a 70° slope angle, 20 m height, and safety berm. The underground mining design includes a 62° ore body dip, crown pillar to prevent surface subsidence, and specific stope dimensions with support from sill pillars and rock bolts. The study analyzes the influence of stope sizes on pit wall and pit bottom stability, observing slope failures near the hanging wall. Overall, the findings demonstrate a successful transition from surface to underground mining at the Sepon Gold mine with stable operations.
This study is important as it provides valuable insights and design considerations for ensuring stability during the transition from surface to underground mining, contributing to safe and efficient mining operations at the Sepon Gold mine.
The introduction section is poor. You must extend it. So in the introduction section of the mentioned research paper, the following issues could be raised:
- The need for a comprehensive assessment of slope stability during the transition from surface to underground mining at the Sepon Gold mine.
- The potential risks and challenges associated with the transition process, including the stability of surface slopes and the design of underground mining methods.
- The importance of understanding the geological characteristics of the ore body and their influence on slope stability and mining operations.
- The significance of employing numerical analysis, such as the Generalized Hoek-Brown criterion, to evaluate slope stability and make informed decisions regarding design and safety measures.
- The relevance of this study's findings for the mining industry, as they provide insights into successful strategies for ensuring stability during the transition from surface to underground mining, which can be applied to similar mining operations.
Also the introductory section of this paper requires improvement in order to better engage and inform the reader about the topic at hand.
Please consider below mentioned papers.
https://doi.org/10.3390/geosciences12050199
https://doi.org/10.33271/mining15.01.050
https://doi.org/10.3390/mining1010008
https://doi.org/10.1007/s11081-017-9361-6
I believe it worth considering in your paper.
In your research authors must discuss below mentioned issues (indicate lines):
- How was the stability of surface slopes assessed prior to commencing underground operations, and what specific factors were considered in the evaluation?
- Can you elaborate on the numerical analysis employed for evaluating slope stability, such as the finite element method? How was the Generalized Hoek-Brown criterion incorporated into the analysis?
- What criteria or requirements were used to determine the pit design, including the slope angle, height, and safety berm dimensions, to ensure stability during surface mining operations?
- In the underground mining design, how was the crown pillar thickness determined to prevent surface subsidence, and what considerations were made regarding its location and dimensions?
- Could you provide more details on the use of rock bolts for support in specific stope areas and the continuous monitoring system for surface deformation? How were the stope sizes determined, and what impact did they have on the stability of pit walls and the pit bottom during underground mining?
Please provide a short description of further research.
The conclusions section of the paper should highlight the novelty and importance of the research to ensure a clear understanding of its original contributions and potential implications for future research and practical applications. Emphasizing the innovative aspects of the study in a concise and effective manner is crucial in providing a comprehensive overview of the research in the concluding section.
Overall, the article presented has left a favorable impression, and with the incorporation of the suggested revisions and considerations put forth, it is highly recommended for publication in the "Mining" journal.
Author Response
Dear reviewer 2,
Thank you very much for your careful and professional suggestions. Your comments have been extremely valuable and helpful in revising and improving our paper. We have carefully studied your comments and made the necessary corrections, hoping they meet with your approval. We have also provided a point-by-point response to address each of your comments as following:
The introduction section is poor. You must extend it. So in the introduction section of the mentioned research paper, the following issues could be raised:
- The need for a comprehensive assessment of slope stability during the transition from surface to underground mining at the Sepon Gold mine.
- The potential risks and challenges associated with the transition process, including the stability of surface slopes and the design of underground mining methods.
- The importance of understanding the geological characteristics of the ore body and their influence on slope stability and mining operations.
- The significance of employing numerical analysis, such as the Generalized Hoek-Brown criterion, to evaluate slope stability and make informed decisions regarding design and safety measures.
- The relevance of this study's findings for the mining industry, as they provide insights into successful strategies for ensuring stability during the transition from surface to underground mining, which can be applied to similar mining operations.
Also, the introductory section of this paper requires improvement in order to better engage and inform the reader about the topic at hand.
Please consider below mentioned papers.
https://doi.org/10.3390/geosciences12050199
https://doi.org/10.33271/mining15.01.050
https://doi.org/10.3390/mining1010008
https://doi.org/10.1007/s11081-017-9361-6
Thank you for your comment. We have revised the manuscript based on your suggestions and added additional content to improve the introduction section, as highlighted in yellow. Additionally, we have read the paper you recommended through the provided link, and we found the information to be highly valuable. It has greatly enhanced our understanding of the topic.
Comment 1: How was the stability of surface slopes assessed prior to commencing underground operations, and what specific factors were considered in the evaluation?
Response 1: We greatly appreciate your valuable questions, as they have helped improve our paper. In response to your query, the stability of surface slopes is typically assessed using various methods prior to commencing underground operations. The evaluation takes into account several specific factors to ensure a comprehensive analysis. The following aspects are commonly considered: geotechnical site investigations, slope geometry and configuration, rock mass characterization, slope stability analysis, and geotechnical parameters and safety factors. Furthermore, these assessments are conducted in accordance with the guidelines for open pit mine design.
Comment 2: Can you elaborate on the numerical analysis employed for evaluating slope stability, such as the finite element method? How was the Generalized Hoek-Brown criterion incorporated into the analysis?
Response 2: Thank you for your comments. We have thoroughly studied your question. In response, the finite element method (FEM) is a numerical technique utilized in rock mechanics to analyze various variables within a rock mass. It involves dividing the rock mass into finite elements or blocks, and equations for each element are solved simultaneously using a digital computer. This approach allows for a comprehensive understanding of the behavior of rock masses under different conditions. In our analysis, we incorporated the Generalized Hoek-Brown criterion as a constitutive model for rock behavior. This criterion establishes a relationship between principal stresses and rock strength parameters. By characterizing the properties of the rock material, calculating rock mass strength, evaluating stress conditions, and comparing calculated strength with applied stress conditions, we verified the stability of the system. Computer programs and software often include the Generalized Hoek-Brown criterion for numerical analysis in geotechnical engineering. These methods and models enable us to gain valuable insights into the behavior and stability of rock masses, contributing to the field of rock mechanics and geotechnical engineering.
Comment 3: What criteria or requirements were used to determine the pit design, including the slope angle, height, and safety berm dimensions, to ensure stability during surface mining operations?
Response 3: Thank you for your comments. Regarding your question, I understand that it can be challenging. Typically, determining the pit design in surface mining operations involves various factors. These include slope angle, height, and safety berm dimensions. The criteria for determining these factors rely on geological and geotechnical assessments, slope stability analyses, safety considerations, and rock mass strength.
The slope angle is chosen to achieve a balance between stability, productivity, safety, and economics. Bench height, on the other hand, is determined based on equipment size and efficiency. Safety berms are constructed to prevent falling material, and ongoing monitoring and maintenance practices are implemented to ensure ongoing stability.
It's important to note that design criteria may vary depending on site-specific conditions and local regulations. We hope that our answer meets the requirements. If you have any further questions, please feel free to ask.
Comment 4: In the underground mining design, how was the crown pillar thickness determined to prevent surface subsidence, and what considerations were made regarding its location and dimensions?
Response 4: Regarding the design in the paper, to prevent crown pillar failure or surface subsidence, it is recommended to have a thickness of not less than 40-50 meters based on the results. Additionally, the factor of safety is considered to ensure underground stability.
Comment 5: Could you provide more details on the use of rock bolts for support in specific stope areas and the continuous monitoring system for surface deformation? How were the stope sizes determined, and what impact did they have on the stability of pit walls and the pit bottom during underground mining?
Response 5: Regarding your question, our design for the open stope focuses on controlling the stope size to minimize the potential for failure. The size of the stope can significantly impact surface slope deformation, with larger dimensions increasing the likelihood of failure. However, in our design, we have made an attempt to avoid the use of rock bolt support in order to optimize the economic aspect.
In the conclusion, I have revised and made changes to the last part of the paper on page seven. I sincerely apologize if my answer is not clear enough or does not meet your requirements. Your comments have provided valuable technical questions that have helped improve my knowledge on the topic.
We tried our best to improve the manuscript and made some changes in the manuscript. These changes will not influence the content and framework of the paper. And here we did not list the changes but marked yellow in revised paper.
We appreciate for your warm work earnestly, and hope that the correction will meet with your approval.
Once again, thank you very much for your comments and suggestions.
Reviewer 3 Report
The article contains interesting numerical studies on the stability of excavations at the transition from the open pit to the underground method. Six different design of open stop dimension were presented in a very interesting way, which were analyzed in terms of displacement, stress ratio and safety factor. The presented research issue is very important due to the continuity of operation while maintaining a high safety factor. Below are some comments and suggestions:
1. In the introduction, it should be mentioned that one of the most important aspects in the transition from the open-pit method to the underground method is the method of accessing the deposit by means of access excavations, which determine not only the method of transporting the output, but also the mining relations between the preparatory and operating excavations (doi:10.3390 /en15228740). In addition, information on the severity of fault analysis methods for heavy equipment and their components used in mining (doi:10.3390/en15176263) should also be added;
2. Figures 5a, 5b, 5c, in particular the numerical values on the vertical and horizontal axes are illegible even when enlarged - they should be corrected;
3. In the third chapter, it should be written whether the modeling took into account the contact between the layers and the shear strength;
4. For Fig. 7, the mining method should be described, in particular the order of deposit extraction, dimensions of excavations and the expected daily progress, excavation support and method of excavation liquidation - these are basic and very important information that affect the stability of mining excavations;
5. In the subsection 4.2, it should be written which of the six variants of the stop design is closest to real conditions (near the Sepon gold mine deposit) and what factors affect the geometry of the excavations and sill pillars;
6. In the fourth chapter concerning the analysis, reference should be made to several literature items in which the impact of the stress ratio and geometry of excavations on their stability was examined - so that the results of the calculations constituted a form of discussion of the results;
7. Several numerical values resulting from numerical modeling should be added to the conclusions, primarily for crown and sill pillars and excavations - as recommendations for mines.
Author Response
Dear reviewer 3,
Thank you very much for your careful and professional suggestions. Your comments have been extremely valuable and helpful in revising and improving our paper. We have carefully studied your comments and made the necessary corrections, hoping they meet with your approval. We have also provided a point-by-point response to address each of your comments as following:
Comment 1: In the introduction, it should be mentioned that one of the most important aspects in the transition from the open-pit method to the underground method is the method of accessing the deposit by means of access excavations, which determine not only the method of transporting the output, but also the mining relations between the preparatory and operating excavations (doi:10.3390 /en15228740). In addition, information on the severity of fault analysis methods for heavy equipment and their components used in mining (doi:10.3390/en15176263) should also be added;
Response 1: We have reviewed the added introduction section on pages 2-3, which has been highlighted in yellow. The purpose of this addition is to provide a comprehensive description of the issues and literature review related to the transition from surface to underground mining, with a specific focus on ensuring stability. We will carefully consider these changes while reviewing the manuscript to ensure the overall coherence and clarity of the content.
Comment 2: Figures 5a, 5b, 5c, in particular the numerical values on the vertical and horizontal axes are illegible even when enlarged - they should be corrected;
Response 2: We have made edits to the manuscript and added a new figure (page 5) to enhance the content and presentation. We believe these changes contribute positively to the overall quality of the paper.
Comment 3: In the third chapter, it should be written whether the modeling took into account the contact between the layers and the shear strength
Response 3: Regarding this question, we do not focus on that particular issue. Instead, our study centers on the implementation of a model based on the geological model. This model employs the finite element method to assess the potential for failure.
Comment 4: For Fig. 7, the mining method should be described, in particular the order of deposit extraction, dimensions of excavations and the expected daily progress, excavation support and method of excavation liquidation - these are basic and very important information that affect the stability of mining excavations;
Response 4: We have created Figure 7 to illustrate the mining consequence, specifically showcasing the open stope extraction process utilizing the overhand cut method. The dimensions of the sill pillar and crown pillar and the bench geometry in the figure has been described above the figure. However, it is important to note that the expected daily progress has not been included in this particular illustration.
Comment 5: In the subsection 4.2, it should be written which of the six variants of the stop design is closest to real conditions (near the Sepon gold mine deposit) and what factors affect the geometry of the excavations and sill pillars;
Response 5: We have reviewed the revised version of the manuscript and have taken note of the edits made to section 4.2.1 on pages 10-11. The changes have been highlighted in yellow.
Comment 6: In the fourth chapter concerning the analysis, reference should be made to several literature items in which the impact of the stress ratio and geometry of excavations on their stability was examined - so that the results of the calculations constituted a form of discussion of the results
Response 6: We have added references to several relevant literature items that impact the results. These references have been included on lines 301 (page 10), 364 (page 12), 400 (page 14), 415 (page 15), and 478 (page 17), and have been highlighted in yellow
Comment 7: Several numerical values resulting from numerical modeling should be added to the conclusions, primarily for crown and sill pillars and excavations - as recommendations for mines.
Response 7: We have modified the conclusion based on the comment provided on the last page. The changes have been made to ensure that the conclusion aligns with the feedback given.
Once again, please allow us to express our thanks to reviewer for your careful and professional comments, which are very helpful to improve our manuscript. All the other Suggestions have been improved accordingly in our paper. All the content modification proposed by the reviewer, we have marked it as red. We tried our best to improve the manuscript and made some changes in the manuscript. These changes will not influence the content and framework of the paper. We appreciate your warm work earnestly and hope that the correction will meet with approval.
Round 2
Reviewer 1 Report
The paper was corrected according my suggestions.
Author Response
Done
Reviewer 2 Report
Good revision.
Author Response
Done